

# Double-Fourier engineering of Josephson energy-phase relationships applied to diodes

A. Mert Bozkurt[1,2*], Jasper Brookman[1], Valla Fatemi[3†] and Anton R. Akhmerov[1‡]

**1** Kavli Institute of Nanoscience, Delft University of Technology,
P.O. Box 4056, 2600 GA Delft, The Netherlands
**2** QuTech, Delft University of Technology, P.O. Box 4056, Delft 2600 GA, The Netherlands
**3** School of Applied and Engineering Physics, Cornell University, Ithaca, NY 14853 USA

⋆ a.mertbozkurt@gmail.com , † vf82@cornell.edu , ‡ superdiode@antonakhmerov.org

## Abstract

We present a systematic method to design arbitrary energy-phase relations using parallel arms of two series Josephson tunnel junctions each. Our approach employs Fourier engineering in the energy-phase relation of each arm and the position of the arms in real space. We demonstrate our method by engineering the energy-phase relation of a near-ideal superconducting diode, which we find to be robust against the imperfections in the design parameters. Finally, we show the versatility of our approach by designing various other energy-phase relations.

See also: Online presentation recording.

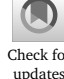

# 1 Introduction

The Josephson tunnel junction is the fundamental building block of superconducting circuits [1]. These junctions have enabled the development of a wide range of functional devices such as superconducting quantum interference devices (SQUIDs), superconducting low-inductance undulatory galvanometers (SLUGs) [2], superconducting nonlinear asymmetric inductive elements (SNAILs) [3,4], quantum-limited amplifiers [5–7], and a bevy of superconducting qubits [8,9].

An example device that can be realized using Josephson junctions is a superconducting diode: a junction with unequal critical currents in different directions. Superconducting diode effect manifests generically in inhomogeneous Josephson junctions subject to a magnetic field [10,11]. Recently, however, there has been renewed interest in studying different physical mechanisms for the creation of superconducting diodes. While superconducting diodes require breaking both time-reversal and inversion symmetries—otherwise the current-phase relationship (CPR) is anti-symmetric in phase—the way in which these symmetries are broken reveals information about the underlying physical systems. To name several examples, recent studies reported superconducting diode effect in spin-orbit coupled in $2d$-electron gases under external magnetic field [12,13], superconducting thin films [14–19], topological insulators [20,21], finite-momentum superconductors [22]. An alternative to controlling the junction CPR for creating a supercurrent diode is to combine multiple junctions in a supercurrent interferometer either consisting of multiple high transparency junctions [23–25] or arrays of Josephson tunnel junctions [26].

We propose a systematic approach to engineer arbitrary energy-phase relationships (EPRs) of a two-terminal device using parallel arrays of Josephson tunnel junctions. We draw inspiration in the observation that circuits of conventional tunnel Josephson junctions implement a variety of Hamiltonians [27–29], originally proposed for difficult-to-engineer microscopic structures. We improve on the results of Ref. [26], which used an exponentially large number of perfectly identical Josephson junctions to create a single Fourier component of the EPR, followed by combining individual Fourier components. We show that the EPR of a Josephson junction array can be engineered by combining Fourier engineering of the EPRs of each arm of the array, variation of the arm strengths in real space, and phase offsets created by an external magnetic field. Our design relies on using standard fabrication techniques and is resilient against fabrication imperfections. We promote that the schemes presented here may be useful in designing sophisticated energy-phase landscapes for decoherence-protected qubit designs [30].

# 2 The arbitrary EPR algorithm

Our conceptual algorithm relies on the following realizations:

- The current-phase relation of two Josephson junctions in series matches the functional form of that of a short Josephson junction with a finite transparency. This allows single-parameter control over the Fourier components in the energy-phase relation of the arm.

- The energy-phase relation of multiple parallel junctions is a convolution of the individual energy-phase relations with the vector of junction strengths when each arm has equal phase offsets and transparency.

- Shifting the total Josephson energy of all arms by the same amount does not change the lowest Fourier components, and therefore the overall shape of the current-phase relation stays the same.

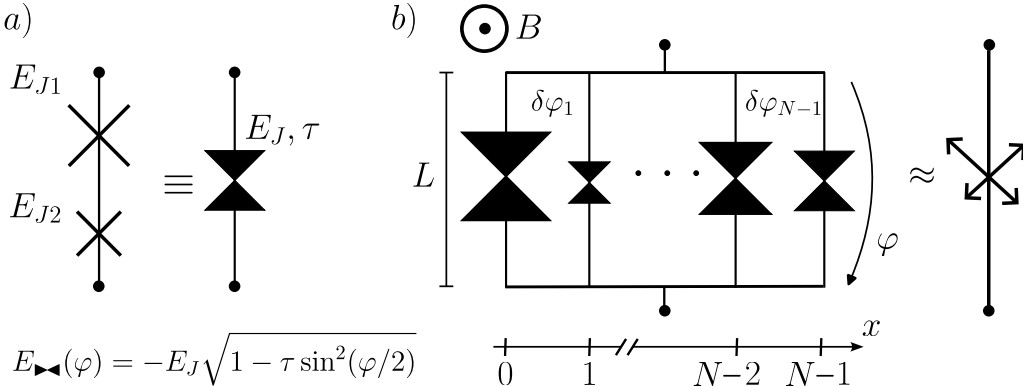

Figure 1: (a) The elementary unit of the design. Two Josephson tunnel junctions in series is equivalent to a short, single channel superconductor-normal metal-superconductor junction. The energy-phase relation describing the arm is given in the equation below. (b) The layout of the Josephson junction array with $N$ arms connected in parallel. Magnetic field $B$ points out of the plane of the Josephson junction array, giving rise to phase difference $\delta\varphi_n = BL(x_n - x_{n-1})$ between arms $n-1$ and $n$, where $x_n$ denotes the position of the $n$-th arm and $L$ is the length of the loop in $y-$direction. We denote the resulting EPR of the Josephson junction array as a two-terminal circuit element.

The elementary unit of the design used to generate higher harmonics of a CPR, of an arm of the Josephson junction array, consists of two Josephson tunnel junctions connected in series, with Josephson energies $E_{J1}$ and $E_{J2}$ [see Fig. 1(a)]. The EPR of each Josephson junction is $U_i(\varphi_i) = -E_{Ji}\cos(\varphi_i)$, where $\varphi_i$ is the phase drop across the junction.[1] We consider the classical limit $E_J \gg E_C$ and neglect the charging energy of the island. A residual charging energy only incrementally changes the CPR [32], which does not qualitatively influence our approach. Current conservation and the additivity of phase differences yields:

$$E_{J1}\sin(\varphi_1) = E_{J2}\sin(\varphi - \varphi_1), \tag{1}$$

where $\varphi = \varphi_1 + \varphi_2$, with $\varphi$ the total phase difference across the arm (see Figure 1). Solving for $\varphi$, we obtain the CPR of an arm:

$$I_{\blacktriangleright\blacktriangleleft}(\varphi) = \frac{E_J \tau}{4\Phi_0} \frac{\sin(\varphi)}{\sqrt{1 - \tau\sin^2(\varphi/2)}}, \tag{2}$$

with $\Phi_0 = \hbar/2e$ the superconducting flux quantum. The corresponding EPR is

$$E_{\blacktriangleright\blacktriangleleft}(\varphi) = -E_J\sqrt{1 - \tau\sin^2(\varphi/2)}, \tag{3}$$

where $E_J \equiv E_{J1} + E_{J2}$ is an overall Josephson energy of an arm and $\tau \equiv 4E_{J1}E_{J2}/(E_{J1} + E_{J2})^2$ controls the relative strength of the higher harmonics of the EPR. This EPR has the same functional form as that of a short, single-channel finite transparency junction with transparency $\tau$ and gap $E_J$—a remarkable coincidence, for which we have no explanation.[2] A Cooper pair transistor also exhibits an identical EPR albeit being in the deep charging regime $E_C \gg E_J$ [35].

---

[1] We ignore the weak higher harmonic terms that have recently been reported in single Josephson tunnel junctions [31]. Their influence can be easily incorporated and does not substantially alter the claims of our work.

[2] While the Eq. (2) is known, see *e.g.* [33,34], to the best of our knowledge, its correspondence with that high of a high transparency short junction was not previously reported.

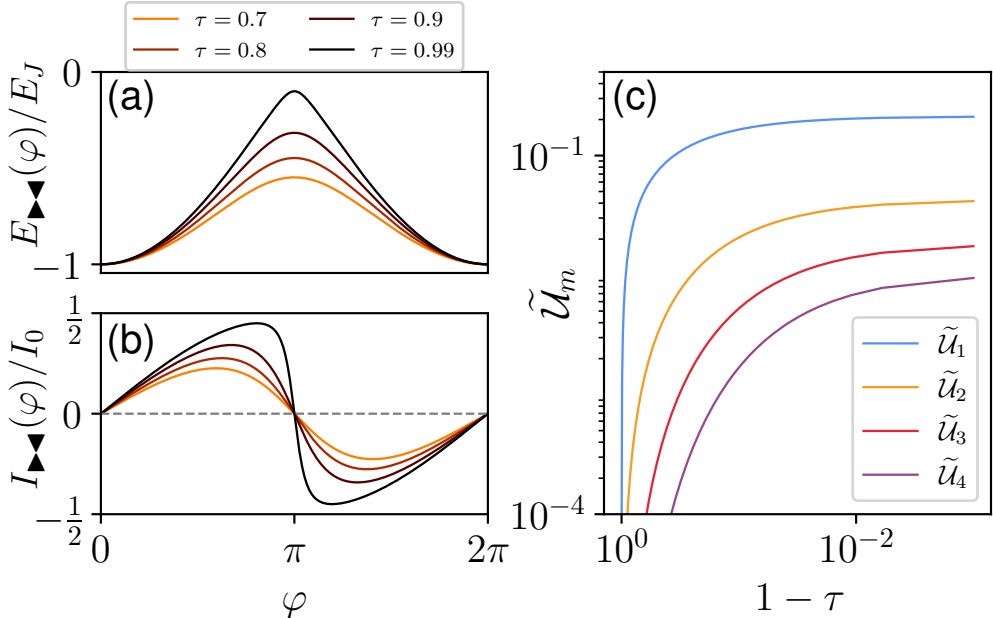

Figure 2: (a) EPR in units of $E_J$ and (b) CPR in units of $I_0 \equiv E_J/\Phi_0$ for a single arm with various values of $\tau$. Higher $\tau$ values introduce higher Fourier components (c).

The EPR and CPR of an arm become highly nonsinusoidal at $\tau \approx 1$ or $E_{J1} \approx E_{J2}$, see Fig 2. We introduce the Fourier transform of the normalized EPR of an arm:

$$\mathcal{U}(\tau, \varphi) = \sqrt{1 - \tau \sin^2(\varphi/2)} \equiv \sum_{m=-\infty}^{\infty} \widetilde{\mathcal{U}}_m(\tau) e^{im\varphi}, \tag{4}$$

where $\widetilde{\mathcal{U}}_m$ are the Fourier coefficients of $\mathcal{U}(\tau, \varphi)$. In the high transparency limit, $\tau \approx 1$, $\widetilde{\mathcal{U}}_m \sim 1/m^2$ for $m \lesssim 1/(1-\tau)$. We plot $\tau$-dependence of several lowest Fourier coefficients of a single arm EPR in Fig. 2(c).

With this way to create higher order harmonics of a single arm EPR, we utilize a Josephson junction array shown in Fig. 1(b) to engineer arbitrary EPRs. In addition to varying the strengths of each Josephson junction, and therefore $E_{J,n}$ and $\tau_n$ of $n$-th arm, we utilize phase offsets by adding magnetic flux between the arms. Magnetic flux gives rise to phase differences $\delta\varphi_n$ between arms $n$ and $n-1$. In this way, we shift the phase offset of each arm by an amount $\phi_n = \sum_{n'=1}^{n} \delta\varphi_{n'}$ with respect to a reference arm $n = 0$. For the rest of the discussion, we define an arm strength distribution by assigning a position to each arm, namely $E_{J,n} \equiv E_J(x_n)$, and correspondingly distributions of the effective transparency $\tau_n \equiv \tau(x_n)$ and phase offsets $\phi_n \equiv \phi(x_n)$.

The EPR of the Josephson junction array is

$$U(\varphi) = -\sum_{n=0}^{N-1} E_J(x_n)\mathcal{U}(\tau(x_n), \varphi + \phi(x_n)), \tag{5}$$

where $N$ is the total number of arms. This EPR is highly nonlinear in $\tau_n$ and $x_n$, and linear in $E_J$. Our goal is to find $U(\varphi)$ that approximates a target EPR, $U_{\text{target}}(\varphi)$, by optimizing the design parameters $E_J$, $\tau$ and $\phi$. Because the role of $\tau$ is to introduce higher harmonics, and the role of $x_n$ is to break time-reversal symmetry, we choose to make $\tau_n$ and $x_n$ uniform to simplify the problem. Specifically, we use $\phi(x_n) = 2\pi n/N$ and $\tau(x_n) = \tau \approx 1$, which makes the right hand side of Eq. (5) a convolution of $E_J(x_n)$ and $\mathcal{U}(\tau, \varphi)$. We then find an

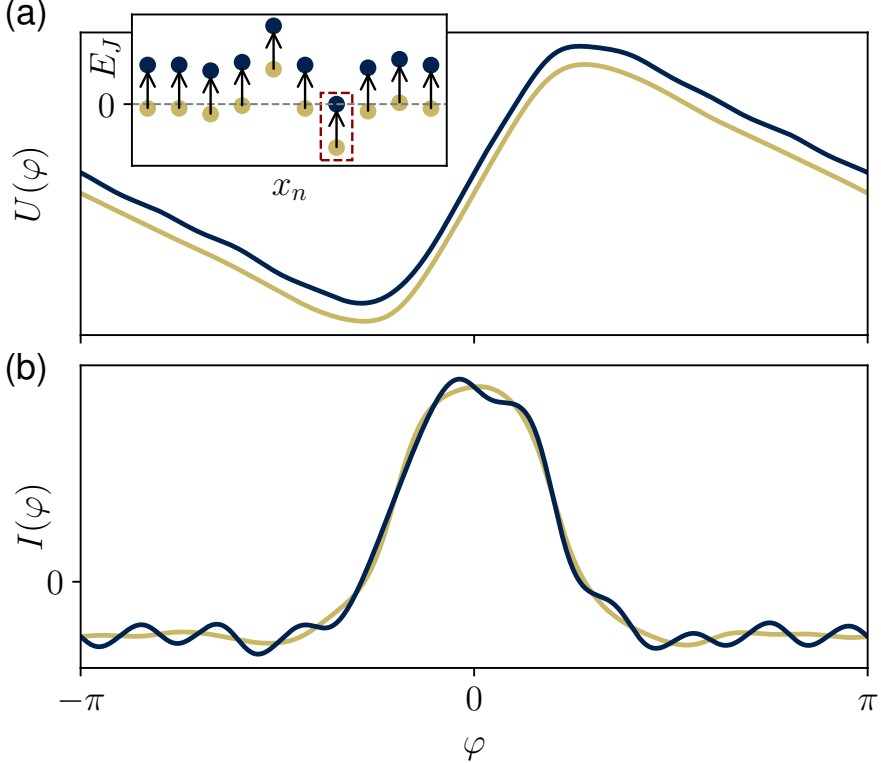

Figure 3: (a) EPR and (b) CPR of two sets of Josephson energies for $N = 10$ arms and $\tau = 0.98$, where yellow lines represent the case with negative junction strengths and blue lines represent the case with non-negative junction strengths. Here, we shift all the Josephson energies by adding the most negative junction strength. Inset shows the Josephson energies distributions of the corresponding EPR and CPR. Here, we shifted the blue EPR vertically for visualization purposes. The red box with dashed lines in the inset shows junction with the most negative strength.

approximate solution of the optimization problem by requiring that two EPRs agree at a set of discrete points $U(2\pi m/N) = U_{\text{target}}(2\pi m/N)$, with integer $0 \le m < N$. In other words, the Josephson junction strengths $E_J(x_n)$ are obtained by Fourier transforming $U_{\text{target}}$, dividing the coefficients by the Fourier components of $\mathcal{U}(\varphi)$ and applying an inverse Fourier transform:

$$E_J(x_n) = -\mathcal{F}^{-1}\left\{ \frac{\mathcal{F}\{U_{\text{target}}(\varphi_m)\}}{\tilde{\mathcal{U}}_m} \right\}_n. \tag{6}$$

In general, the set of Josephson energies $E_J(x_n)$ found by inverse discrete Fourier transform includes negative values, whereas the stable state of a single arm has a positive $E_J$. We resolve this obstacle by adding the most negative $E_{J,\text{min}}$ to all the Josephson energies $E_J(x_n)$. Because $\sum_n \mathcal{U}(\phi - 2\pi n/N)$ has a period of $2\pi/N$, $N$ of its lowest Fourier components are absent, and therefore adding it to the EPR only changes it minimally, as shown in Fig. 3. This concludes the design of a Josephson junction array with a target EPR.

## 3  Optimizing the superconducting diode efficiency

We now apply our approach to design a superconducting diode. This device has an asymmetric CPR with unequal critical currents in opposite directions. The diode efficiency $\eta$ is the degree

of asymmetry of its two critical currents:

$$\eta = \frac{|I_{c+} - I_{c-}|}{I_{c+} + I_{c-}}, \tag{7}$$

where $I_{c\pm}$ are the maximum critical currents for current flow in opposite directions. An ideal superconducting diode with $\eta = 1$ has a sawtooth-shaped EPR:

$$U_{\text{sawtooth}}(\varphi) = \frac{\varphi}{2\pi} - \left\lfloor \frac{\varphi}{2\pi} \right\rfloor, \tag{8}$$

where $\lfloor \varphi \rfloor$ is the floor function.

To optimize a superconducting diode, we apply the algorithm of the previous section with $U_{\text{target}} = U_{\text{sawtooth}}$, with the results shown in Fig. 4. Because $U_{\text{sawtooth}}$ is discontinuous, its Fourier approximation exhibits oscillatory behavior near the discontinuity, known as the Gibbs phenomenon. This reduces the superconducting diode efficiency by allowing small side peaks of the opposite sign next to the main peak in the CPR. To attenuate the Gibbs phenomenon, we modify the Fourier coefficients of $E_J$ using the $\sigma$-approximation [36]. In Fig. 4(a), we demonstrate the effect of the $\sigma$-approximation on CPR of a superconducting diode. With increasing degree of regularization the efficiency of the superconducting diode increases and eventually peaks at $\eta = 0.92$ (for $N = 78$ arms). We then choose a degree of regularization

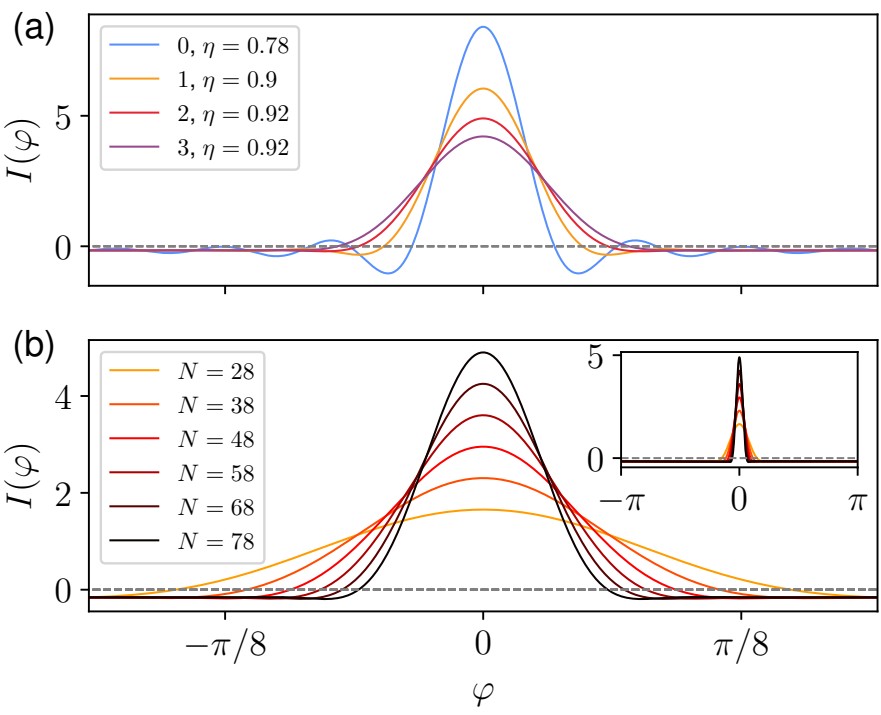

Figure 4: Superconducting diode CPR. Panel (a) shows CPR for a superconducting diode with $N = 78$ arms for different degrees of $\sigma$-regularization. We mark the degree of regularization and the resulting efficiency in the legend. (b) CPR for a superconducting diode constructed with several different number of arms for $\tau = 0.95$. With increasing $N$, main peak in the CPR gets narrower and higher. Inset shows the overall phase range of the CPR.

that maximizes the efficiency for a given number of arms and $\tau$. In Fig. 4(b), we show $N$ dependency of the EPR and CPR of a Josephson junction array for a fixed $\tau = 0.95$. As $N$ increases, the main peak in the CPR gets higher and narrower, resulting in a larger efficiency.

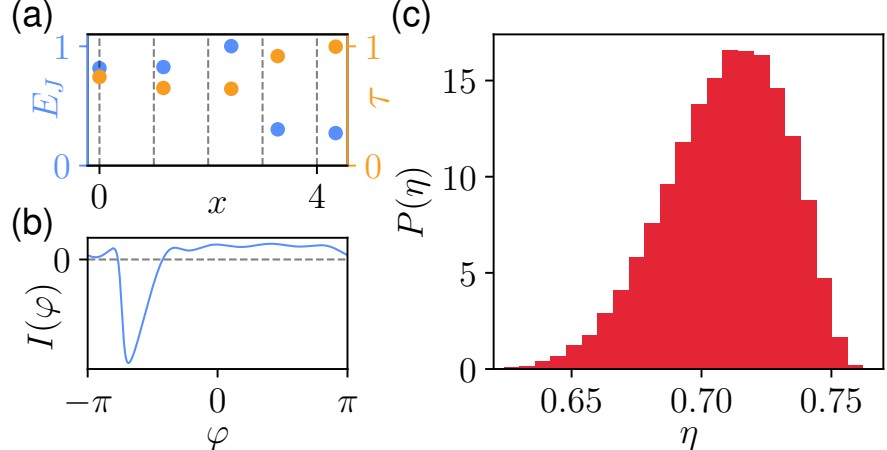

Figure 5: Stochastic optimization for $N = 5$ arms and the effect of presence of disorder on diode efficiency. In (a), we show the optimized arm positions, $E_J$ (blue), and $\tau$ (yellow) for each arm. For convenience, left y-axis represents $E_J$ whereas right y-axis represents $\tau$. The dashed gray line represents the equally spaced arm positions. (b) shows the resulting CPR of the Josephson junction array shown in (a). In (c), we display the probability distribution of efficiency for 50000 disorder realizations in junction strength. Here, we renormalized the histogram such that area under the histogram integrates to 1.

## 4 Generalization of the algorithm

The discrete Fourier transform approach yields a closed form solution, it applies to any target EPR using the setup of Fig. 1. On the other hand, it relies on several simplifications:

- It makes $U(\varphi)$ agree with $U_{\text{target}}(\varphi)$ at $N$ points, instead of minimizing an error norm.

- It requires that all $\tau_n$ are equal and $x_n$ are equidistant.

- It does not take into account the random variation of junction strengths.

To relax the first limitation we observe that as long as the error norm is quadratic in $U(\varphi) - U_{\text{target}}(\varphi)$, the optimization problem stays a least squares problem (LS), implemented in the SciPy library [37]. Relaxing the second and third limitations makes the problem nonlinear, but keeps it solvable using stochastic global optimization techniques.

To apply LS to the superconducting diode design, we use the error norm

$$\min_{E_J(x_n)} \sum_i \left[ U''(\varphi_i) \right]^2 , \tag{9}$$

which makes the negative current as constant as possible in the range $\varphi_i \in [\varphi_{\min}, \varphi_{\max}]$, with $\varphi_{min}$ and $\varphi_{max}$ being the upper and lower boundary values for the phase range used for applying LS method. To make the solution nonzero we fix $E_J(x_0) = 1$ and solve for the Josephson junction strengths of the remaining $N - 1$ arms. After finding a solution to the LS problem, we add the most negative junction strength, similar to the Fourier method. We then apply a brute force optimization to determine the phase region $[\varphi_{\min}, \varphi_{\max}]$ that yields highest $\eta$.

We solve the nonlinear problem by applying the SciPy's [37] implementation of the differential evolution method [38] to the problem of finding $\max_{\{x_n\}, \{E_{Jn}\}} \eta$ for a given $N$. This

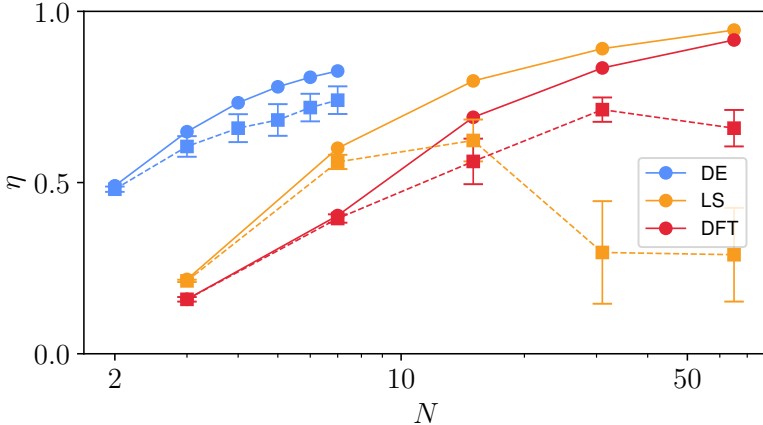

Figure 6: Comparison of the diode efficiencies generated by the three optimization methods: differential evolution method (blue), LS method (orange) and discrete Fourier transform (red). Solid lines and circular markers represent the ideal disorder-free limit. Dashed lines and square markers represent the disordered limit with a margin of ±2% randomness in junction parameters. The error bars are used to demonstrate the standard deviation of the efficiency distribution that arises from the presence of disorder.

procedure yields the results shown in Fig. 5. Because differential evolution allows the presence of noise, we allow the junction strengths to vary by ±2%, similar to the experimental state of the art [39]. We find mean diode efficiency of $\eta \approx 0.71$ for $N = 5$ arms, much larger than the result of the Fourier method.

In Fig. 6, we compare the diode efficiencies produced by the three optimization methods in perfect conditions and in presence of noise. All three methods show improvement with increasing $N$. We observe that both the Fourier and the LS approaches become more sensititve to disorder with increasing $N$. This happens because the typical $E_J$ of each arm is comparable to the maximal one due to the shifting by the minimal value. This results in the root-mean-square variation $\sim \sqrt{N}$ not being suppressed with $N$. The differential evolution method yields highest efficiencies for low $N$ and converges the fastest, while showing only limited degradation in presence of disorder. The superior performance of this method is expected, however the computational costs become prohibitively high for large $N$. The discrete Fourier transform method is the most constrained, and therefore it performs worst, albeit the difference with LS vanishes at high $N$. The LS approach is the least resilient to disorder once $N$ becomes large due to overfitting.

# 5   Other example EPRs

To demonstrate the generality of our approach, we apply it to other example EPRs: a square wave, a triangular wave, and a double well potential. For square and triangular wave potentials, we employ the discrete Fourier transform approach. Similar to the superconducting diode EPR case, we choose a constant $\tau$ and solve for the Josephson energy distribution. The convergence of this method with $N$, shown in Fig. 7 confirms that it allows to generate arbitrary EPRs. The double well EPR example demonstrates how to apply the same device to design an EPR that is only defined within a limited phase range. Specifically, we consider a

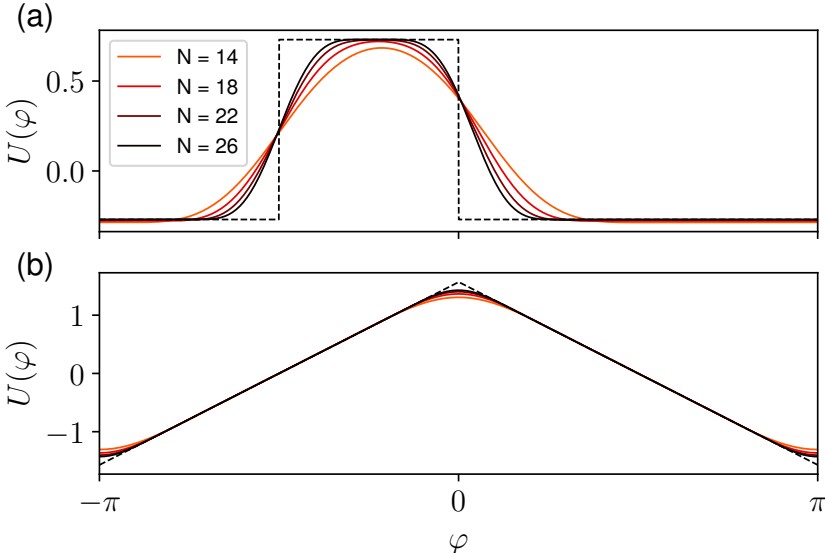

Figure 7: (a) Square wave and (b) triangular wave EPRs for various $N$. The black dashed lines depict the target EPR for each case. For visualization purposes, we subtract the mean from each EPR. For simulating sharp features of the target EPR, we choose $\tau = 0.97$ to include higher Fourier components.

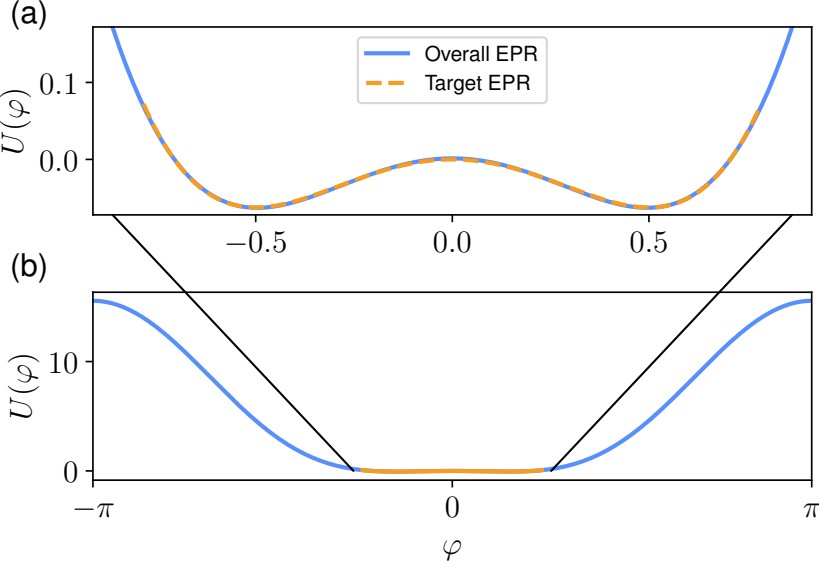

Figure 8: Engineering a double well potential using a Josephson junction array with $N = 4$ arms and $\tau = 0.1$. Panel (a) displays the EPR within the phase region of interest, whereas panel (b) shows the entire phase range. The yellow dashed line depicts the target EPR, $U_{\mathrm{dw}}(\varphi)$ given in Eq. (10), in the phase range of interest. The blue line depicts the full EPR of the Josephson array.

double well potential of the form:

$$U_{\mathrm{dw}}(\varphi) = \varphi^4 - \frac{1}{2}\varphi^2. \tag{10}$$

By discretizing Eq. (5) and eliminating equations outside the region of interest, we obtain an overdetermined set of equations, which we solve using LS and shift the Josephson energies by the most negative one when necessary. Due to absence of sharp features in double well potential, we choose a low value of $\tau = 0.1$. The resulting EPR of the Josephson junction array with $N = 4$ arms, shown in Fig. 8, agrees with target EPR given in Eq. (10) in the phase region of interest, depicted by the yellow dashed line.

## 6 Conclusion and outlook

We proposed and investigated an approach to design arbitrary energy-phase relationships using Josephson tunnel junction arrays. In particular, our approach allows to design a superconducting diode with a desired efficiency and the resulting design is robust against variation in device parameters.

The main building block of our approach is possibly the simplest source of a non-sinusoidal CPR: two Josephson tunnel junctions in series. While our method does not rely on a specific arm EPR, this choice offers practical advantages. For example, more than two junctions in series generally have a multi-valued CPR [29] and does not allow for a simple parametrization. An alternative way of generating higher harmonics is a Josephson junction in series with an inductor [40, 41], however it has a non-periodic CPR, and is therefore more complicated to use.

We have focused on the DC properties of the circuit, and we envision engineering the RF characteristic as the next logical step. For example, we expect that diode effects are correlated with odd-order RF nonlinearities, which we could explore [4]. Furthermore, so far, we have ignored the role of junction capacitance $E_C$, which sets the plasma frequency of the superconducting junctions, and consequently the islands. This plasma frequency limits the range of operation frequencies, therefore incorporating the dynamics of the superconducting islands into the picture would be relevant for designing quantum coherent devices. Finally, we can extend our scheme to 2- or 3-dimensional energy-phase landscapes and include sensitivity to parametric knobs as optimization inputs for design of protected qubits [30].

## Acknowledgments

We acknowledge useful discussions with Alessandro Miano, Nicholas E. Frattini, Pavel D. Kurilovich, Vladislav D. Kurilovich, and Lukas Splitthoff.

**Author contributions** A.R.A. and V.F. defined the research question. A.R.A oversaw the project. J.B. implemented the initial version of the optimization as a part of his bachelor project. A.M.B. implemented the final version of the optimization and performed the numerical simulations in the manuscript. All authors contributed to identifying the final algorithm. A.M.B., A.R.A. and V.F. wrote the manuscript.

**Funding information** This work was supported by the Netherlands Organization for Scientific Research (NWO/OCW) as part of the Frontiers of Nanoscience program, an Starting Grant 638760, a subsidy for top consortia for knowledge and innovation (TKl toeslag) and a

NWO VIDI Grant (016.Vidi.189.180). AMB acknowledges NWO (HOTNANO) for the research funding.

**Data availability** The code used to generate the figures is available on Zenodo [42].

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
