# Peer review of "Double-Fourier engineering of Josephson energy-phase relationships applied to diodes"

_SciPost Physics, doi:SciPost Phys. 15, 204 (2023)_

## Round 1 · Referee Report · Jean-Damien Pillet (Referee 1) · 2023-8-19

Strengths

1) The authors present several methods to create arbitrary energy-phase relationships U(\phi) using networks of tunnel Josephson junctions. 2) The theoretical proposal appears to be generally feasible experimentally. 3) The idea of using the higher harmonics of the energy-phase relations of two Josephson junctions in series as a resource to engineer an arbitrary potential is very ingenious. 4) The example of the Josephson diode effect is very illustrative and quite spectacular. The presented method allows for achieving nearly perfect efficiencies. 5) Their methods are also employed to generate potentials of square, triangular, and double-well shapes. In fact, this is a very modern issue because the creation of certain precise Josephson potentials theoretically enables the generation of protected qubits, as explained in reference 30.

Weaknesses

1) The article emphasizes the realization of a Josephson diode without necessarily explaining the motivation behind it. Nevertheless, this is a very trendy topic and will likely interest many readers. 2) I somewhat regret that the method is not applied beyond the one-dimensional case. I would have greatly appreciated seeing an example of how this ingenious approach optimizes the design of a protected qubit. I guess the authors are already working on it and another manuscript about that will come in the future. 3) The basic building block enabling the realization of the circuits proposed by the authors consists of two tunnel Josephson junctions in series. This forms a superconducting island (between the junctions) whose charge can in principle influence the device's behavior. The authors do not discuss this aspect at all. For Josephson energies E_J much larger than the charge energies E_C of the junctions, the presence of this island is insignificant, and the authors are right to disregard it. However, for small junctions, this doesn't appear legitimate. This should be mentionned. 4) Generating energy-phase relationships with sharp features requires having high "transmissions" (as defined in the manuscript). In the case where the role of the charge of the island becomes relevant, achieving this doesn't seem straightforward. I will elaborate on this point further in my report.

Report

This manuscript presents theoretical work in which the authors have conceived several ingenious methods to create Josephson potentials U(\phi) of arbitrary shapes using circuits composed solely of tunnel Josephson junctions. This is, of course, interesting for multiple reasons, such as the realization of Josephson diodes or potentials that enable the creation of a protected qubit (see reference 30). The title of the manuscript, "Double-Fourier engineering of Josephson energy-phase relationships applied to diodes," actually highlights only one of these methods, which is based on a double Fourier transform. This particular method is nice as it provides an analytical expression for the Josephson energies of each junction. Nonetheless, the other methods (LS method and differential evolution method), which are more numerical, yield better results.

I enjoyed reading this manuscript a lot. The work is interesting, of excellent quality, and the manuscript's writing is very clear. One of the central predictions of this manuscript is, in fact, that the energy-phase relation (EPR) of two tunnel Josephson junctions connected in series is the same as that of a single junction but with high transparency. Hence, an effective transparency \tau can be defined, which, as explained by the authors, is 1 for identical junctions and 0 for highly asymmetric Josephson energies E_J1 >> E_J2. Cases of high \tau allow for obtaining higher-order harmonics in the EPR and realizing Josephson potentials of much more varied shapes than the simple cosine of a single tunnel junction, simply by assembling these basic units in parallel. I do not think this prediction is truly novel; after all, this building block is a Cooper pair transistor, as studied for many years. Nevertheless, it is employed in a new, intelligent and experimentally feasible strategy to design arbitrary Josephson potential.

Overall, I only have positive comments about this manuscript. Therefore, I am in favor of its publication.

However, I would like to make a comment on a point in the manuscript that surprises me a bit: the role of the charge energy on the DC properties of the proposed devices is not mentioned at all. This might be for a reason I have completely missed during my reading. Yet, the basic building block (two tunnel Josephson junctions in series) constitutes a Cooper pair transistor with a central island whose charge energy could play a crucial role.
In the limit E_J >> E_C, the presence of this island is irrelevant and can be neglected as the authors have done. Otherwise, if the charge energy E_C is not negligible compared to the Josephson energy E_J, then the charge of this island can be controlled with a gate voltage, greatly affecting the energy-phase relationship of the device. This is detailed, for example, in the thesis of Philippe Joyez from the Quantronics group (available at https://iramis.cea.fr/spec/Pres/Quantro/static/publications/phd-theses/index.html).

On page 59 of this reference, the following energy-phase relationship is derived for two identical tunnel Josephson junctions in series. It is (up to a constant):

E(\phi) = -E(0) (1 - \tau sin^2(\phi/2))^(1/2)

where E(0) = (E_J^2 + 4E_C^2 (1 - n_g)^2)^(1/2) and \tau = E_J^2 / E(0)^2. Here, n_g is the gate charge controlled by a gate voltage. It is similar to expression 3 of the manuscript. This expression is obtained within a two-band model where the Hilbert space is reduced to just two states (0 or 1 Cooper pair in the island) and is valid under the conditions E_C >> E_J and 0 < n_g < 2. It is clear that if n_g is not equal to 1, then the transmission is less than 1 and can even be close to 0 if E_C is much larger than E_J. In this case, E(\phi) is proportional to cos(\phi), i.e., an EPR without higher harmonics. This implies that the method proposed by the authors cannot be applied and it is not possible to engineer sharp features in the EPR. To connect this with the manuscript, I imagine that for two junctions with different Josephson energies, an expression for the transmission similar to a Breit-Wigner form might emerge:

\tau ~ (4E_J1 * E_J2) / ((E_J1 + E_J2)^2 + 16E_C^2 (1 - n_g)^2)

where the transmission is controlled not only by the ratio between Josephson energies but also by the charging energy. Of course, one could consider choosing all n_g values to be 1, but this would require having as many gates as islands, which is a significant experimental challenge if N is comparable to or larger than 100.

For this reason, I believe it might be interesting for the readers to mention in the manuscript that their work is valid in the case of E_J >> E_C, and maybe say a few short words on the role of E_C in the DC properties of the device.

I hope the authors will find these comments useful and that they will help to improve the manuscript.

Requested changes

I don't believe that this manuscript requires any major changes to be published; it is interesting, clear, and well-written. However, I can still suggest a few minor corrections, the relevance of which I leave to the authors' judgment. 1) Discuss briefly the role of the charging energy for the DC properties of the device. Possibly explain that the method presented is only valid for E_J much larger than E_C. 2) Define \phi_min and \phi_max more explicitly. 3) The caption of figure 7 should start with "(a)" instead of "(Engineering a)". 4) In the caption of figure 8, (a) and (b) are not explicitly mentionned.

---

## Round 1 · Referee Report · Anonymous (Referee 2) · 2023-8-22

Report

The authors of the manuscript under consideration start with the remarkable observation that two Josephson junction in series feature a current-phase relation that resembles that of a single channel quantum point contact. Using such an element as building block they investigate how to engineer quasi arbitrary current-phase relations by a parallel configuration of a large number of those blocks with numerically determined parameters. By adding a flux through the ring form by subsequent blocks they can generate a diode effect. Several ways to optimize the diode efficiency are investigated numerically and efficiencies of almost unity can be achieved. Several other examples are discussed.

The article is well written, treats an interesting and timely subject and will likely make an impact both in future experimental Josephson devices as well in further theoretical developments. Hence, the manuscript should be considered further by SciPost once a few comments below are addressed.

Requested changes

1) In fact, a recent preprint (ref. [26]) makes a rather similar observation and comes to similar conclusions. It would be fair to comment in the present manuscript on the relation to that work. E.g. the procedure to maximize the diode efficiency are almost the same.

2) In Fig. 6 it is quite remarkable that disorder seems to have a strongly negative effect on the diode efficiency for large junction numbers. Maybe the authors can comment on this kind of counter-intuitive behavior.

---

## Round 3 · Referee Report · Jean-Damien Pillet (Referee 1) · 2023-9-21

Report

The authors have made a few minor changes to the manuscript following feedback from the two referees. The article is still excellent. Therefore, I have not changed my mind and I have nothing to add since my last report.

I think the manuscript can be accepted as is.

---

## Round 3 · Author Response

We thank both referees for their overall positive evaluation. We have implemented all the requested changes. We also provide the redlined manuscript with the changes highlighted at this URL: https://surfdrive.surf.nl/files/index.php/s/NwTnLoOVPoXTTGD

Ref 1:

One of the central predictions of this manuscript is, in fact, that the energy-phase relation (EPR) of two tunnel Josephson junctions connected in series is the same as that of a single junction but with high transparency. Hence, an effective transparency \tau can be defined, which, as explained by the authors, is 1 for identical junctions and 0 for highly asymmetric Josephson energies E_J1 >> E_J2. Cases of high \tau allow for obtaining higher-order harmonics in the EPR and realizing Josephson potentials of much more varied shapes than the simple cosine of a single tunnel junction, simply by assembling these basic units in parallel. I do not think this prediction is truly novel; after all, this building block is a Cooper pair transistor, as studied for many years. Nevertheless, it is employed in a new, intelligent and experimentally feasible strategy to design arbitrary Josephson potential.

We agree with the referee that the current-phase relation of two Josephson tunnel junctions is not truly novel, and indeed it has been studied for many years. We would like to clarify, though, that we consider a classical circuit $E_J \gg E_C$, and therefore not a Cooper pair transistor. While the current-phase relation of two tunnel junctions in series was studied in e.g. our Ref. 32, to the best of our knowledge, its correspondence to that of an SNS junction was not stated explicitly. The referee is also correct in saying that the main focus of our work is utilizing this current-phase relation to construct arbitrary ones.

For this reason, I believe it might be interesting for the readers to mention in the manuscript that their work is valid in the case of $E_J \gg E_C$, and maybe say a few short words on the role of $E_C$ in the DC properties of the device.

1) Discuss briefly the role of the charging energy for the DC properties of the device. Possibly explain that the method presented is only valid for E_J much larger than E_C.

Following the referee's suggestion we have now: - Explicitly stated that we neglect the charging energy when introducing the setup and stated that we expect that the residual charging energy will not qualitatively affect our conclusions. - Added a statement that also a Cooper pair transistor has the same current-phase relation to the discussion of current-phase relation correspondence.

2) Define $\phi_{min}$ and $\phi_{max}$ more explicitly.

We now explicitly define these in text.

3) The caption of figure 7 should start with "(a)" instead of "(Engineering a)". 4) In the caption of figure 8, (a) and (b) are not explicitly mentionned.

We thank the referee for pointing these out, and we have updated the captions accordingly.

Ref 2.

1) In fact, a recent preprint (ref. [26]) makes a rather similar observation and comes to similar conclusions. It would be fair to comment in the present manuscript on the relation to that work. E.g. the procedure to maximize the diode efficiency are almost the same.

We thank the referee for their remark. We have added a sentence to the introduction, clarifying the relation of our work to Ref. 26.

2) In Fig. 6 it is quite remarkable that disorder seems to have a strongly negative effect on the diode efficiency for large junction numbers. Maybe the authors can comment on this kind of counter-intuitive behavior

We now added a statement to the manuscript explaining that the increased disorder sensitivity for large junction numbers is due to unsuppressed root-mean-square fluctuations of the junction strengths.

---

## Editorial Decision

published